# Therapy Used to Promote Disease Remission Targeting Gut Dysbiosis, in UC Patients with Active Disease

**DOI:** 10.3390/jcm11247472

**Published:** 2022-12-16

**Authors:** Hengameh Chloè Mirsepasi-Lauridsen

**Affiliations:** 1Health Gut Inn Balance APS, Kloevermarken 7, 2680 Solroed Strand, Denmark; hcl@gut-in-balance.dk

**Keywords:** ulcerative colitis, fecal microbiota transplantation, immunosuppressant, TNF-α blocker, probiotic, prebiotic, VSL #3, *E. coli* Nissle 1917

## Abstract

Ulcerative colitis (UC) is a relapsing non-transmural chronic inflammatory disease of the colon characterized by bloody diarrhea. The etiology of UC is unknown. The goal is to reduce the inflammation and induce disease remission in UC patients with active disease. The aim of this study is to investigate the innovative treatment method used to promote disease remission in UC patients with active disease targeting gut dysbiosis. Immunosuppressants such as TNF-α blocker are used to promote disease remission in UC, but it is expensive and with side effects. Probiotic, prebiotic and diet are shown to be effective in maintaining disease remission. Fecal microbiota transplantation (FMT) might be the future therapy option to promote disease remission in UC patients with active disease. However, correct manufacturing and administration of the FMT are essential to achieve successful outcome. A few cohorts with FMT capsules show promising results in UC patients with active disease. However, randomized controlled clinical trials with long-term treatment and follow-up periods are necessary to show FMT capsules’ efficacy to promote disease remission in UC patients.

## 1. Introduction

Inflammatory Bowel Disease (IBD) is divided into Crohn’s Disease and Ulcerative Colitis (UC). Ulcerative colitis (UC) is a relapsing non-transmural chronic inflammatory disease that is restricted to the colon, and during flares the disease is characterized by bloody diarrhea. Worldwide prevalence of UC has been reported with 156 to 291 cases per 100,000 persons per year, and these numbers are increasing by 2% each year [1].

The etiology of the UC is unknown, but studies indicate several possible causes such as nutrition [2,3], bacteria [4,5], virus [6,7], environmental factors, and host immune systems [8,9]. The gut microbiota of UC patients contributes to initiation and/or maintenance of the inflammatory states by providing antigens or co-stimulatory factors that drive the immune response in a misdirection in these genetically susceptible hosts [8,10]. UC pathogenesis is linked to alteration in the makeup of the intestinal microbiota, including a reduced diversity of intestinal microbiota species/dysbiosis in comparison to non-UC people [11]. Bacteriological analysis of stool from UC patients shows increased prevalence of Enterobacteriaceae- and Morganellaceae-rich communities compared to the non-UC person [12]. Studies indicate increased prevalence of virulent *Escherichia coli* (*E. coli*) species belonging to Extraintestinal Pathogenic *E. coli* (ExPEC), *Klebsiella and Citrobacter* in UC patients with active disease [13].

Animal model studies indicate that UC most likely arises because of poor regulatory T-cell function, leading to the overproduction of proinflammatory cytokines such as tumor necrosis factor alpha (TNF-α) and interleukin-12 (IL-12) [14]. Impaired production or function of known regulatory/immunosuppressive cytokines such as IL-10 is also linked to UC pathogenesis [8].

There is no cure for UC, and the only treatment is to reduce the inflammation in patients with disease relapses using anti-inflammatory medicine. Promoting remission in UC patients with active disease is very challenging and expensive [15]. In the last decade, the most-used medication in UC patients was biological medicine, which is very expensive and has many side effects such as liver disease, cancer, Lupus-like syndrome, heart disease, and central nervous system disease [16,17]. Biological medicine consists of antibodies that block specific parts of the immune system such as TNF-α blockers, reducing inflammation [18].

As mentioned earlier, dysbiosis in UC patients might be the trigger of disease relapses. Intestinal microbiota therapy might promote disease remission in UC patients. Diet, probiotic, prebiotic, and fecal Microbiota transplantation (FMT) is a way of promoting symbiosis in the intestines of UC patients. In the next few sections, the therapeutic effect of diet, Probiotic, prebiotic, and FMT will be discussed.

## 2. UC Pathogenesis

### 2.1. Environmental Factors and Diet

Studies based on genetic variety in UC patients did not show any significant gene-variety linked to UC, but Human Leukocytes antigen (HLA) class II genes DR2 and DR9 are linked to UC, while DRB1*0103 is significantly associated with disease susceptibility and increased risk of colectomy [19]. TNF-α is identified as a susceptibility locus for UC. Therefore, biological agents as anti-TNF- α antibodies/TNF- α antagonists are used to neutralize TNF-α and reduce the inflammation in UC patients with severe disease. Reduced production of Melatonin hormone is linked to UC [20], which might be the cause of depression and anxiety in UC patients [21,22]. Melatonin in GI tract is produced by enterochromaffin cells (EC), which are located in epithelial layer of GIT [21]. Melatonin has an immunoregulatory effect, exerts antimicrobial action and modulates microbial components [23]. Melatonin reduces certain opportunistic pathogen bacterial genera such as Desulfovibrio [15], Peptococcaceae, and increases the abundance of beneficial genera such as Bifidobacterium [22]. Melatonin upregulates the major mucin MUC2 to maintain intestinal integrity [24]. Therefore, Melatonin therapy might be beneficial in UC patients.

Environmental factors and diet have been linked to UC pathogenesis [25]. Improvements in personal hygiene and reduced exposure to microbial stimulation through human contact in the Western countries reduces immune tolerance and promotes autoimmune diseases such as UC. Dietary changes and increased usage of antibiotics alter intestinal microbiota and microbial mediated mechanisms of immunological tolerance [26].

There is a link between cesarean birth and UC, as the neonate will be inoculated with hospital bacteria instead of the mother’s bacteria through the birth canal [27]. Studies show high-fat/high-sugar diets cause microbial dysbiosis, decrease mucus layer thickness, and increase intestinal permeability and susceptibility [28]. Dietary carbohydrates such as starches and fibers ferment to “short-chain fatty acid” (SCFA), such as acetate, propionate, and butyrate [29]. SCFA serves as an inhibitor of proinflammatory cytokine expression in the intestinal mucosa, as well as a stimulator of mucin and antimicrobial peptide production and a strengthener of epithelial barrier integrity by increasing the expression of tight junction (TJ) proteins [30]. A decreased prevalence of butyrate-producing bacteria, such as *Clostridiales* species, in UC patients with active disease explains the decreased amount of SCFA in UC patients’ fecal samples [31]. The studies show that nutritional therapy with prebiotic properties enables modulation of the gut microbiota and regulation of the immune defense in UC, and promotes mucosal healing [32,33]. Fecal Microbiota Transplantation in combination with Anti-inflammatory diet is shown to be effective in promoting disease remission in UC patients with active disease [34]. Kedia et al. 2022 show that long-term anti-inflammatory diet is more effective to maintain disease remission in UC patients in comparison to standard medical therapy [34].

It is also important to mention the neuroimmunological point of view when evaluating the factors that might promote gut dysbiosis in UC. In the last two decades the focus has been on the “microbiota–intestine–brain” axis, how gut dysbiosis alters production of serotonin, dopamine, GABA and noradrenaline, and the capabilities of generating or worsening directly related psychopathologies such as anxiety, depression, mood disorders, schizophrenia, psychotic and personality disorder. Increased risk of psychiatric disorders including suicide attempts is linked to IBD, and stress is known to be a trigger of disease relapse [35]. Studies indicate that psychological stresses such as circadian disruption, sleep deprivation [36], environmental extremes (high altitude, heat, and cold), environmental pathogens, toxicants, pollutants, noise [37], and extreme physical activity [37] have biological effects in hosts such as modulation of gut microbiota toward dysbiosis. Circadian rhythms [38] are clock-controlled genes such as light–dark cycles, shift work, rotating work, feeding schedules and diet composition [39]. Studies show that disruption of Circadian rhythms increases gut permeability [40], alters immune responses [41], increases susceptibility to inflammation [42], causes gastrointestinal epithelium damage and promotes UC [43].

### 2.2. Dysbiosis in UC Patient

Dysbiosis plays an essential role in UC patients. Antibiotic therapy using ciprofloxacin or rifaximin in UC patients with active disease is shown to be effective in promoting disease remission [44]. Increased prevalence of virulent *E. coli* species from B2 phylogenetic groups, which is linked to ExPEC, has been reported in UC patients with active disease (Figure 1). UC associated *E. coli* harboring alpha hemolysin gene is shown to damage epithelial TJ in heterogeneous human epithelial colorectal adenocarcinoma (Caco-_2_) cells [45], and cause extra-intestinal infection in UC mouse model (dextran sulfate sodium treated) [46,47]. A microbiome study of UC patients with active disease has shown decreased residence of anaerobic bacteria such as *Faecalibacterium prausinitzii*, belonging to Clostridial cluster IV [31], which is a butyrate-producing bactrium with anti-inflammatory properties that promotes gut health. An increase in potentially pathogenic bacteria such as *Clostridiodes difficile* [48], *Salmonella* spp. [49] *Listeria monocytogenes* [50], *Klebsiella*, *Enterobacter, fusobacterium* [51], *proteus* spp. and some viruses [52] is reported in UC patients in comparison to healthy persons [53,54]. Peptococcaceae and Enterobacter species can be pathogentic and cause opportunistic infection in immunocompromised patients, and *Fusobacterium* is linked to colon cancer [51]. A significant reduction in *lactobacilli* and *Bifidobacterium* is reported in UC patients [55] (Figure 1).

## 3. The Effect of Probiotic and Prebiotic in UC Patients

The efficiency of probiotic microorganisms depends on several factors such as their viability, stability during processing and storage, and manufacturing process, causing survival of probiotic through the stomach acidic environment to the intestine where it can finally colonize [56]. Prebiotic combined with probiotic enhances the survival and efficacy of probiotic to influence the composition of the intestinal microbiota and alter the metabolic properties of the microbiome. Prebiotic consists of *Psyllium,* oligofructose-enriched insulin, dietary carbohydrate, starches and fibers, which can be fermented to SCFA by intestinal anaerobic bacteria, and decrease intraluminal PH, making the disease intestinal environment suitable for probiotic to adapt [57,58]. Physiological and microbial changes in the gastrointestinal of UC patients need to be addressed when treating these patients with probiotic. UC patients exhibit minor PH elevation in the small intestinal, where PH is 6 in healthy people while it is 7.5 in the small intestinal of UC patients (Figure 1). In healthy persons the large intestinal PH is 6.7, while in the UC patients’ large intestinal PH varies between 2.3 and 5.5 [59] (Figure 1). The changes in the intestinal PH of UC patients causes intestinal microbiota displacement, such as increased prevalence of *E. coli* spp. in the small intestine, which usually colonizes the large intestine and decreases prevalence of *Clostridiales* spp. [31], which increased in healthy persons small intestine producing SCFA [59] (Figure 1). Increased *Bacteroidetes*, *Peptostreptococcus* and *Eubacteria* [55] are also reported in the colon of UC patients, while in the colon of healthy persons increased prevalence of *Firmicutes*, *proteobacteria* and *Actinobacteria* are reported. Studies indicate decreased food orocecal transit times in the UC patients caused by small intestinal bacterial overgrowth [60] (Figure 1).

Alteration in composition of GIT microbiota in UC patients followed by changes in intestine motility and PH promotes dysbiosis. The most-used probiotics in UC patients are *E. coli* Nissle 1917 and VSL#3, which mainly consist of *lactobacilli* and *Bifidobacterium* spp. and *streptococcus* [56]. *E. coli* Nissle 1917 belongs to the B2 phylogenetic group, which is mostly associated with ExPEC. In vivo studies indicate that *E. coli* Nissle 1917 has immunoregulatory properties such as decreasing the number of T cells within the intestinal mucosa, as well as reducing the secretion of proinflammatory cytokines while stimulating the secretion of regulatory proteins such as IL 10 [61,62,63]. However, *E. coli* Nissle 1917, from the B2 phylogenetic group [47], harbors genes for colibactin synthesis inducing DNA damage which is linked to colorectal cancer [64].

There are few randomized controlled clinical trial studies in UC patients, suggesting *E. coli* Nissle 1917 is as effective as mesalazine to promote disease remission in UC patients [65,66,67]. The study by Kruis et al. [65,66] was performed in UC patients with inactive disease using *E. coli* Nissle 1917 as an add-on treatment, while a study by Rembackan et al. [67] was performed in UC patients with active disease. However, in Rembacken et al.’s study corticosteroids were used as add-on treatment, which diminishes the signs/symptoms of any infection/inflammation that might be caused using *E. coli* Nissle 1917. Yet, a randomized double-blind study of *E. coli* Nissle 1917 given as add-on treatment to patients with active UC showed that fewer patients treated with *E. coli* Nissle 1917 had symptomatic remission and that they withdrew from the study. Considering these studies, a larger study is needed to confirm the beneficial effect of *E. coli* Nissle 1917 in UC [68]. The studies of VSL#3 in the UC patients with inactive disease with 6 months follow up show that VSL #3 is suitable to maintain disease remission in UC patients [69,70]. However, longer follow-up time is necessary to demonstrate disease remission in UC patients with inactive disease. Most probiotica products such as VSL#3 consist of *lactobacilli* and *Bifidobacterium* spp. [56]. Lactic acid bacteria such as *lactobacilli* produce bioactive peptides known as bacteriocins that possess antimicrobial activity against pathogenic bacteria [71]. Lactic acid bacteria have immunoregulatory effects and contribute to intestinal host defenses through their interaction with the immune system [72,73]. The Bifidobacterium belongs to the Actinobacteria phylum, which is increased in the breast-fed infants and plays key roles in the maturation of the immune system. Nutrients such as milk oligosaccharides are important drivers of bifidobacterial development. Studies indicate bifidobacteria residing in the intestine of adult humans such as *B. adolescentis* does not utilize milk components, and instead their metabolism complex carbonhydrates in adult-type diet [74]. Bifidobacteria inhibit pathogens through production of organic acids [74], antibacterial peptides [75], by inhibiting quorum-sensing and by stimulating immunity, which are all important to prevent certain infections [74].

## 4. What Is Fecal Microbiota Transplantation?

The benefits of natural microbial compositions were used to cure disease since the 4th century Before Christ. The use of healthy donor feces (Fecal Microbiota Transplantation) as a therapeutic agent for food poisoning and diarrhea was first recorded in the *Chinese Handbook of Emergency Medicine* [76]. Fecal microbiota transplantation (FMT) comprises beneficial bacteria, viruses, parasites, antimicrobial peptides, hormones such as melatonin produced by a healthy gut, metabolic products produced by beneficial bacteria such as SCFA, and nutrition necessary for beneficial microbiota to survive in the disease, etc. FMT consists of all beneficial substances produced by a healthy gut, which can improve a diseased intestinal environment similar to the donor and promote symbiosis. The Challenge with FMT is finding healthy stool donors and regularly screening for pathogenic microorganisms to avoid the risk of infection. Therefore, it is very important to follow the international guidelines regarding donor banking and donor screening [77,78,79,80].

FMT enables remission in the disease intestine, by providing all the beneficial substances missing and by changing the disease intestinal environment such as by changing intestinal PH [81,82,83,84]. When manufacturing FMT products it is essential to be mindful of all above-mentioned beneficial substances in the FMT. Over 50% of intestinal microbiota are obligate anaerobes and cannot live at normal atmospheric concentration [85]. When investigating the FMT manufacturing processes in some published studies, most of the studies produce their FMT products under non-anaerobic conditions; on the contrary, they add more air to their FMT product by using non-anaerobic media and blender to mix the stool [86]. Some studies add water as a diluent when manufacturing FMT products [87]. Water is hypotonic, and since concentration of the solute is greater inside the bacteria water will penetrate the bacterial cell. Some studies produce FMT capsules by diluting 50 gr. of stool with 500 mL of solution, mixing and centrifuging the raw material over 20 min, discharging the supernatant and using the pellet in FMT capsules [88]. The discharged supernatant is as effective as pellets, as it contains beneficial products such as SCFA, etc. Few studies use freeze-dried methods to produce FMT capsules [89]. A freeze-dried method is widely used to preserve single bacteria in early stationary phase/probiotic, but the process is not ideal to preserve viruses and other beneficial substances in the FMT [90,91]. Freeze-dried method is optimized for pressure and temperature for a single bacterium in the stationary phase. However, when freeze drying FMT it is impossible to optimize the pressure and temperature, so it is optimal for all beneficial substances and microorganisms in the FMT. Furthermore, dilution and centrifugation processes in freeze-dried method and regent, used in such as skimmed milk, might be harmful to some of the beneficial substances and bacteria in FMT in donor stool.

As a result, only some of the microorganisms and substances can survive the freeze-drying process, followed by reduced product effects. When using cryoprotectants such as glycerol to prevent formation of ice crystals on the bacterial cell membrane, it is essential to be mindful of host wellbeing. As cryoprotectant is known to increase the human cell-osmolarity followed by intestinal-cell-damages. By adding 30% [88] of cryoprotectant in the final FMT product, it will damage the host intestinal tissue in the long term [92,93,94]. Ting Zhang et al. (2019) recommends “wash microbiota transplantation (WMT)” in order to reduce the toxicity of the FMT, by reducing the number of the white blood cells in the final FMT product [95]. However, many centrifuging and washing steps used in the process harm the beneficial substances in the donor stool, especially anaerobic microbiota, and reduce the efficacy of the final WMT product [69]. Gently mixing and filtering of donor stool in an anaerobic environment is essential to remove the food leftovers and large particles. However, a well-screened donor without eggs or parasites shows no side effects in published studies [80]. More studies are needed in order to investigate the effect of FMT in comparison to WFT.

It is essential to consider before FMT treatment if the patient needs the whole profile of FMT or intermediate parts of the beneficial substance of FMT, followed by adjusting the FMT manufacturing process for that purpose [96,97]. The intermediate part of the beneficial substance of FMT products filtered from bacteria is effective to some extent in promoting healing in the intestines of *Clostridiodes difficile* infected patients [98]. Dysbiosis in *Clostridiodes difficile* infected patients is often caused when using long-term antibiotics or when the patient is immunosuppressed, caused by other chronic disease or in the elderly [80]. When treating chronic inflammatory diseases such as UC patients with active disease to reestablish intestinal symbiosis and induce remission, it is essential to provide a whole profile of FMT/all beneficial substances in FMT to change the disease intestine environment and microbiota. FMT administration methods vary depending on disease. When treating *Clostridiodes difficile* infection, often 1 or 2 FMT treatments using FMT capsules, or liquid by endoscopy or enema should be enough to reestablish intestinal microbiota symbiosis [80]. However, when treating patients with chronic disease/systemic disease, long-term treatment with FMT is necessary since its efficacy is time limited as seen when treating with biological medication [99]. The goal is to reduce the inflammation, induce and maintain remission in patients with active UC disease. Endoscopy is the golden standard of FMT administration methods in which the FMT will be released in the inflammation region in the colon, but there are complications and increased costs linked to endososcopy. When treating patients with FMT it is important to be mindful of the region of inflammation in the disease gut and how to reach the inflammation area to promote disease remission (Figure 1). As seen in Figure 1, when treating inflammation in the small intestine, FMT capsule or FMT given by nasojejunal tube are effective. When treating inflammation in the colon area, FMT capsule or FMT liquid given using endoscopy, colonic transendoscopic enteral tubing (TET) [100] or by nasojejunal tube are effective. However, when treating inflammation around the rectal area, enema is the best way to reach the inflammation region and promote disease remission in UC patients with active disease. Colonic TET is another way of administrating FMT to the colon region, which requires cleansing of the intestine, intravenous anesthesia followed by several endoscopies, where TET inserted into the ileocecal junction is fastened to distal colon and affixed to the skin of the buttocks with the valve connected to the terminal TET [100]. TET is an innovative way to administer FMT in UC patients who need longer-term FMT treatment. However, studies show that 70% of patients with TET had 7 days of retention time within the colonic lumen, where tubes are falling out spontaneously [101].

FMT capsule is the most effective way of treating intestinal dysbiosis/inflammation in patients with chronic disease such as UC, who regularly need treatment with FMT over a longer time. FMT capsules reach out to the gastrointestinal tract of the patient (Figure 1) and can be handled by the patients at their home, which reduces the cost to the healthcare sector and is less demanding for the patients. Combination therapy might be necessary in UC patients with increased inflammation in the rectum, using FMT capsule combined with enema (Figure 1). There are a few reasons for limited FMT-capsules therapy in healthcare sectors: (A) cost prohibitive—researchers cannot afford access to it when one portion of FMT capsule for oral administration costs US dollars 2050; (B) error in FMT manufacturing and administration followed by misleading research published in international papers; (C) doubts regarding FMT safety—however, as with blood, if the FMT international or national guidelines are met FMT is as safe as blood and tissue given to patients in need [77,78,79,80]. In the next section, the effect of FMT to promote disease remission will be discussed.

### Fecal Microbiota Transplantation Treatment in UC Patients

Many cohorts have been published on FMT treatment in UC patients [86]. However, only a few randomized and controlled clinical trial studies have been performed. In the randomized controlled clinical trial by Paul Moayyedi et al. [87] 75 UC patients with active disease were included, and as add-on treatment patients were treated with 50 mL of FMT or placebo retention enema once per week over 6 weeks with 12 months follow up (Table 1). However, only 24% of the patients treated with FMT achieved clinical remission in comparison to 5% in the placebo group [87].

In the study by Noortje et al. [102] 50 UC patients with active disease were included in the study, treated with nasojejunal tube twice at the start of the study and 3 weeks later, with 12 weeks follow up. The study outcome shows 41% disease remission in patients treated with FMT, versus 25% of the patients treated with autologous FMT achieving clinical remission at week 6. However, there were no significant differences after a 12-week follow up [102]. The study by Costello et al. [103] includes 64 UC patients with active disease, using autologous FMT or stool pooled from 3 or 4 donors. The FMT was administered as 50 gr. stool in 200 mL saline/glycerol using colonoscopy and twice of 25 gr. stool in 100 mL saline/glycerol enema was administered in the following 7 days with 12 months follow up. The study outcome shows 55% clinical remission in FMT-treated patients versus 23% clinical remission in autologous FMT after 8 weeks, and 42% of the FMT treated patients were still in clinical remission after 12 months [103].

A randomized controlled clinical trial study by Sood et al. [104] includes 61 UC patients with inactive disease, 30 patients receiving placebo and 31 patients receiving FMT by colonoscopy once a week for 48 weeks. FMT treatment was prepared with 100 gr. of donor stool mixed with 200 mL of saline. The study outcome shows 27 of 31 patients treated with FMT continue maintaining clinical remission, while 18 of the patients also show endoscopic remission versus 8 of 30 patients in the placebo group who showed endoscopic remission [104]. However, this study was based on 112 UC patients with active disease treated with FMT by colonoscopy, of whom 65 achieved clinical remission and after 8 weeks were included in the study by Sood et al. [104] (Table 1).

There are limited randomized clinical trial studies investigating the effect of FMT capsule in patients with UC. However, few cohort studies are published such as the one by Steube et al. [105] including 8 UC patients with active disease. Patients were treated with FMT capsules twice a day with 5 capsules each time for 12 weeks. The study outcome shows 7 of 8 patients achieved clinical improvement, while 5 of them achieved improvement of Mayo Endoscopic subscore. As study limitation, there is no information of the stool dosage given to the patients and there are limited patients included in the study.

The study by Cold et al. [80] includes 7 UC patients with active disease, treated with 50 gr. stool of multi-donor FMT capsules daily for 50 days as add-on treatment with 6 months of follow up. A total of 5 of the 7 patients with active disease achieved remission in weeks 4–8 when treated with FMT capsules. However, 3 of the 5 UC patients in remission had disease relapses after the end of treatment. There are limited studies on long-term FMT-capsule therapy in UC patients. Table 2 shows that 68 FMT clinical trials in UC are registered in clinicaltrial.gov, of which only 56 of them completed or are ongoing. Most of the registered studies are based on FMT administered using endoscopy, enema or via nasojejunal tube. More studies are needed using FMT capsules to treat patients with ulcerative colitis.

## 5. Discussion

The etiology of UC is unknown, but evidence indicates that intestinal microbiota plays an essential role in disease relapses. Animal model discovery indicates that germ-free animals generally do not develop intestinal inflammation, and that it requires a certain genetic background to promote intestinal inflammation [106]. Antibiotic therapy and immunosuppressants such as biological medication are shown to be effective in promoting disease remission in UC [33,99]. However, it is important to be mindful, when using antibiotic therapy to promote disease remission in UC patients, to limit antibiotic resistance organisms [99]. Many life-threatening side effects were reported when using biological medication and immunosuppressant [16]. Therefore, patient follow up is important when using immunosuppressant medication. Biological medication is also known to be very expensive, which is why limited usage is advised in the healthcare sector [17].

Dysbiosis is linked to UC disease relapses with limited prevalence of anaerobic bacteria such as Clostridial cluster IV, which is responsible for fermenting dietary carbohydrates, starches, and fibers to SCFA [29]. Increased prevalence of virulent *E. coli* species is linked to UC disease relapse [13,107]. Nutrition therapy has been shown to be more effective than corticosteroids for healing the mucosa [33], which indicates that prebiotic properties enable modulation of the gut microbiota and regulation of the immune defense in UC. Prebiotic in combination with probiotic might maintain disease remission in UC patients by modulating the intestinal environment of UC patients, by reducing small intestinal PH and increasing the large intestinal PH and by optimizing intestinal motility [57,58].

Physiology and microbial changes in the gastrointestinal of UC patients need to be addressed, when treating UC patients with probiotics. PH changes in the intestinal of UC patients, in comparison to the healthy persons, causes intestinal microbiota displacement [60]. Probiotics, which often consist of lactobacillus acidophilus spp., colonizing the small intestinal and thriving best at PH of 5.5–6.5, will not be able to colonize the small intestinal of the UC patients with a PH of 7.5 [108] (Figure 1). Therefore, prebiotic treatment in combination with probiotic is essential to modify the UC intestinal environment suitable for probiotics to colonize. However, studies show probiotic and prebiotic treatment has not been effective to promote disease remission in UC patients with active disease.

FMT is shown as an innovative way of promoting disease remission in UC patients with active disease. However, it is essential to be mindful of the method used to manufacture and administer FMT. The studies indicate long-term treatment with FMT, administered by endoscopy, is effective to promote and maintain UC disease remission [71,72,73,74]. However, endoscopy is an expensive and demanding administration method both for the healthcare sector and for patients in the form of bowel cleansing, etc. [108]. FMT therapy using FMT capsules is another way of effectively treating UC patients with active disease, which enables inexpensive long-term treatment at patients’ homes, changing UC patients’ small and large intestinal environment and microbiota and promoting symbiosis [109]. Randomized controlled clinical trials in UC patients with active disease are necessary to show FMT-capsule efficacy to promote and maintain disease remission. However, to access cheaper and effective FMT products it is essential that FMT production is handled by the experts in hospital using tissue ACT as blood transfusion and tissue transplant. Hospitals need to follow international guidelines for organisation of FMT and donor recruitment as for blood and tissue transplantation, to ensure product safety. Suitable stool donors should be mainly recruited from blood donors/national Blood Donor Corps, who have the necessary knowledge of being donors and are healthy. Hospitals have the necessary experts to examine the stool donor for safety and to select the patient, who might have benefits of FMT therapy.

## 6. Patents

Hengameh Chloe Lauridsen. WO2021130182A1-FMT Capsule, International patent. (2021).

Hengameh Chloe Lauridsen. WO2021130181A1-Novel faecal composition, International patent. (2021).

## Figures and Tables

**Figure 1 jcm-11-07472-f001:**
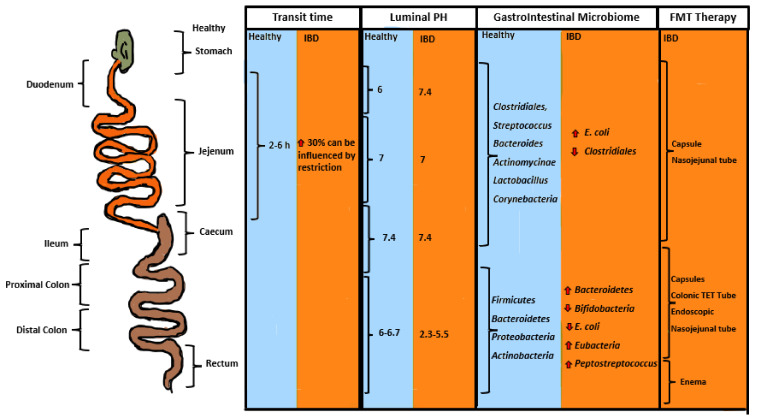
**The first column** shows the bowel transit time in patients with Inflammatory Bowel Disease (IBD)/UC in comparison with healthy people. As shown, the transit time is delayed by 30% in IBD patients. **The second column** shows luminal PH in IBD patients in comparison to healthy people. As shown, the PH is raised in duodenum/jejunum and significantly decreased in the colon of IBD patients in comparison to healthy people. **The third column** shows gastrointestinal microbiome in a healthy person in comparison to IBD patients. Increased prevalence of *E. coli* is seen in the small intestine and absent in colon in comparison to the healthy person. Absence of *Lactobacillus*, *Bifidobacteria*, *Firmicutes*, *Clostridiates* and *Actinobacteria* is also shown in the IBD patients. **The fourth column** shows suggested guidelines for FMT administration to achieve successful FMT therapy.

**Table 1 jcm-11-07472-t001:** Cohort Studies of fecal microbiota transplant for ulcerative colitis: protocol descriptions. Simple Clinical Colitis Activity Index: SCCAI.

Authors	n	Disease Severity	UC Medication	Bowel Preparation	Pre-Antibiotic	FMT Administration
Paul Moayyedi et al. [74]	75	Mayo ≥ 4	Immunosuppressive	None	None	Retention enema
Noortje et al. [91]	48	SCCAI 4-11	None	None	None	Duodenal infusions
Costello et al. [92]	73	Mayo 3-10	Immunosuppressive	None	None	Colonoscopy
Sood et al. [93]	61	Mayo ≥ 2	Mesalazin	None	None	Colonoscopy
Steube et al. [94]	10	Mayo ≥ 4	Immunosuppressive	None	Yes	Capsule
Cold et al. [80]	7	SCCAI 4-10	Standard treatment	None	None	Capsule

**Table 2 jcm-11-07472-t002:** Fecal microbiota transplant in ulcerative colitis resisted in clinicaltrial.gov.

Nr. Clinical Studies	Status	Locations
15	Completed or recruiting	USA
8	Completed or recruiting	Canada
1	Completed or recruiting	Ukraine
2	Completed or recruiting	France
3	Completed or recruiting	Denmark
6	Completed or recruiting	China
7	Completed or recruiting	Netherland
8	Completed or recruiting	Israel
1	Completed or recruiting	Australia
1	Completed or recruiting	Czechia
3	Completed or recruiting	Finland
1	Completed or recruiting	Hong Kong

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
