# Peer review of "Therapy Used to Promote Disease Remission Targeting Gut Dysbiosis, in UC Patients with Active Disease"

_jcm, 2022, doi:10.3390/jcm11247472_

Round 1
Reviewer 1 Report
This is a comprehensive paper worth reading. What needs to be mentioned is that in the field of FMT treatment of UC, Professor Zhang Faming from China has made many valuable contributions, and it is suggested to quote relevant research papers. In addition, the methodology of FMT is a key factor affecting the efficacy and safety of UC treatment. Please briefly elaborate on the washed microbiota transplantation (WMT).
Author Response
Reviewer 1
Q1) This is a comprehensive paper worth reading. What needs to be mentioned is that in the field of FMT treatment of UC, Professor Zhang Faming from China has made many valuable contributions, and it is suggested to quote relevant research papers. In addition, the methodology of FMT is a key factor affecting the efficacy and safety of UC treatment. Please briefly elaborate on the washed microbiota transplantation (WMT).
A1) Thank you for your feedback and suggestion. More about WMT is now added to the manuscript in the section” What is Fecal Microbiota Transplantation” Line 198-203.
Reviewer 2 Report
The manuscript is titled "Therapy used to promote disease remission targeting Gut Dysbiosis, in UC patients with active disease" but discusses the use of fecal microbiota transplantation and the use of capsules to achieve a successful outcome. Revise the title of the article.
Pathogenesis of UC. Is there an effect of hormones in ulcerative colitis? For example, Zhao ZX et al. investigated the relationship between melatonin concentration and severity of ulcerative colitis, revealing a significant negative correlation (Zhao ZX, Yuan X, Cui YY, Liu J, Shen J, Jin BY, Feng BC, Zhai YJ, Zheng MQ, Kou GJ, Zhou RC, Li LX, Zuo XL, Li SY, Li YQ. Melatonin Mitigates Oxazolone-Induced Colitis in Microbiota-Dependent Manner. Front Immunol. 2022 Jan 18;12:783806. doi: 10.3389/fimmu.2021.783806).
The role of dysbiosis is not clear in the manuscript. Why is it a trigger for ulcerative colitis? Bifidobacteria and lactobacilli are only mentioned in the section " Dysbiosis in UC patients", although their significance and protective mechanisms are needed in the section "The effect of probiotic and prebiotic in UC patients ". The effect of probiotics discussed with an example the E. coli Nissle 1917. However, Nissle 1917 hosts in its genome the pks pathogenicity island that codes for the biosynthesis of the genotoxin colibactin. Colibactin is a potent DNA alkylator, suspected to play a role in colorectal cancer development (Nougayrède JP, Chagneau CV, Motta JP, Bossuet-Greif N, Belloy M, Taieb F, Gratadoux JJ, Thomas M, Langella P, Oswald E. A Toxic Friend: Genotoxic and Mutagenic Activity of the Probiotic Strain Escherichia coli Nissle 1917. mSphere. 2021 Aug 25;6(4):e0062421. doi: 10.1128/mSphere.00624-21).
The application of fecal microbiota transplantation has many questions. What microbiota criteria should be applied in fecal microbiota transplantation? Should strains be sequenced? To study their pathogenic properties? To be determined by virome? Metabolic characterization of FMT strains? These are more important questions that determine the quality of FMT and long-term consequences for the patient. These issues need to be discussed in the manuscript.
Author Response
Reviewer 2
Q1) The manuscript is titled "Therapy used to promote disease remission targeting Gut Dysbiosis, in UC patients with active disease" but discusses the use of fecal microbiota transplantation and the use of capsules to achieve a successful outcome. Revise the title of the article.
A1) Thank you for your suggestion. The paper also discusses the effect of probiotic, prebiotic and diet to promote disease remission in UC patients. More studies on effect of diet in UC disease remission is now added to the manuscript section “Environmental factors and diet” line 77-80.
Q2) Pathogenesis of UC. Is there an effect of hormones in ulcerative colitis? For example, Zhao ZX et al. investigated the relationship between melatonin concentration and severity of ulcerative colitis, revealing a significant negative correlation (Zhao ZX, Yuan X, Cui YY, Liu J, Shen J, Jin BY, Feng BC, Zhai YJ, Zheng MQ, Kou GJ, Zhou RC, Li LX, Zuo XL, Li SY, Li YQ. Melatonin Mitigates Oxazolone-Induced Colitis in Microbiota-Dependent Manner. Front Immunol. 2022 Jan 18;12:783806. doi: 10.3389/fimmu.2021.783806).
A2) Thank you for your suggestion. The effect of melatonin is now discussed in the manuscript section “UC pathogenesis” line 61-67 and mentioned in the section “What is Fecal Microbiota Transplantation” line 159.
Q3) The role of dysbiosis is not clear in the manuscript. Why is it a trigger for ulcerative colitis? Bifidobacteria and lactobacilli are only mentioned in the section " Dysbiosis in UC patients", although their significance and protective mechanisms are needed in the section "The effect of probiotic and prebiotic in UC patients ". The effect of probiotics discussed with an example the E. coli Nissle 1917. However, Nissle 1917 hosts in its genome the pks pathogenicity island that codes for the biosynthesis of the genotoxin colibactin. Colibactin is a potent DNA alkylator, suspected to play a role in colorectal cancer development (Nougayrède JP, Chagneau CV, Motta JP, Bossuet-Greif N, Belloy M, Taieb F, Gratadoux JJ, Thomas M, Langella P, Oswald E. A Toxic Friend: Genotoxic and Mutagenic Activity of the Probiotic Strain Escherichia coli Nissle 1917. mSphere. 2021 Aug 25;6(4):e0062421. doi: 10.1128/mSphere.00624-21).
A3) Thank you for your comments. More detail regarding the dysbiosis in UC is now added to the section “Dysbiosis in UC patient” line 82-84 in the manuscript. The role of the colibactin in E. coli Nissle is also added to the manuscript line 118-120. More information about protective mechanisms of probiotic Bifidobacteria and lactobacilli is also added in the section "The effect of probiotic and prebiotic in UC patients " line 128-137.
Q4) The application of fecal microbiota transplantation has many questions. What microbiota criteria should be applied in fecal microbiota transplantation? Should strains be sequenced? To study their pathogenic properties? To be determined by virome? Metabolic characterization of FMT strains? These are more important questions that determine the quality of FMT and long-term consequences for the patient. These issues need to be discussed in the manuscript.
A4) Thank you for your question. Few lines regarding challenges when using FMT is now added to the section” What is Fecal Microbiota Transplantation” line 138-147. Danish national guidelines for FMT screening and FMT biobank is used in Denmark, which shown no side effects in patients treated with FMT products (the reference is added to the manuscript) in our studies. As for treatment consequences for the patients treated with FMT in long-term, so far, the studies published have op till one-year FMT therapy with patient follow up. More research is needed on this subject to analyze the FMT therapy outcome in patient treated for more than a year.
Reviewer 3 Report
The submitted manuscript focuses on potential treatment options for ulcerative colitis remission. Besides probiotics and prebiotics, FMT represents an emerging trend in gut microbiota modulations.
Comment and recommendations to improve the scientific quality and comprehensiveness of the manuscript:
- I suggest checking English throughout the whole manuscript and significantly improving sentence constructions. I would strongly suggest that a native speaker will go through the manuscript and correct it. Please, unify the style of keywords used and check their terminology according to the manuscript.
- It will be beneficial to check singular and plural forms in the sentences and correct English grammar (e.g. Animal model studies indicates,...). Please, replace the Environment factor with Environmental factors.
- Please, check the Italic font in the manuscript and use it uniformly (e.g. spp. and spp.) In addition, use Clostridium difficile, not clostridium.
- I strongly recommend adding 1-2 figures or schemes to the manuscript. It would improve the visualization and readability of the topic for the readers.
- Please, add a section about healthy gut microbiota composition and function, and compare microbiota in health and disease.
- As mentioned in the manuscript, probiotics might serve as microbiota modulators for UC patients. I recommend adding data about their functions and potential mechanisms.
- Please, describe the general advantages and disadvantages of FMT and probiotics.
- I suggest adding a table with ongoing clinical trials focused on the role of FMT or probiotics in UC patients to highlight the clinical relevance of this emerging microbiome-based approach.
Author Response
Reviewer 3
Comments and Suggestions for Authors
The submitted manuscript focuses on potential treatment options for ulcerative colitis remission. Besides probiotics and prebiotics, FMT represents an emerging trend in gut microbiota modulations. Comment and recommendations to improve the scientific quality and comprehensiveness of the manuscript:
Q1) I suggest checking English throughout the whole manuscript and significantly improving sentence constructions. I would strongly suggest that a native speaker will go through the manuscript and correct it. Please, unify the style of keywords used and check their terminology according to the manuscript.
A1) Thank you for your suggestion. A native English speaker have now read and correct the manuscript for English.
Q2) It will be beneficial to check singular and plural forms in the sentences and correct English grammar (e.g. Animal model studies indicates,...). Please, replace the Environment factor with Environmental factors.
A2) Thank you for your suggestion. I have now corrected the plural forms in the sentences.
Q3) Please, check the Italic font in the manuscript and use it uniformly (e.g. spp. and spp.) In addition, use Clostridium difficile, not clostridium.
A3) Thank you for your correction. Clostridium and spp It is now corrected in the manuscript to Clostridium difficile and spp.
Q4) I strongly recommend adding 1-2 figures or schemes to the manuscript. It would improve the visualization and readability of the topic for the readers.
A4) Thank you for your recommendation. One figure is now added to the manuscript.
Q5) Please, add a section about healthy gut microbiota composition and function, and compare microbiota in health and disease.
A5) Thank you for your suggestion. Figure 1 is now added to the manuscript with information about gut microbiota in health and in Inflammatory Bowel Disease and section 2.2 and section 3 address healthy gut bacteria and probiotic in health and disease.
Q6) As mentioned in the manuscript, probiotics might serve as microbiota modulators for UC patients. I recommend adding data about their functions and potential mechanisms.
A6) Thank you for your suggestion. Information regarding the function and potential mechanisms/ effects of probiotics is now added in the manuscript, section "The effect of probiotic and prebiotic in UC patients " line 128-137.
Q7) Please, describe the general advantages and disadvantages of FMT and probiotics.
A7) Thank you for your comments. The advantages and disadvantages of probiotic is described now in more detail in the section “The effect of probiotic and prebiotic in UC patients”. There is no published data regarding disadvantages with VSL #3 probiotic, but the disadvantages when using E. coli Nissle 1917 is now described in this section with more detail. More data regarding advantages and disadvantages, when using FMT is described in the section “What is Fecal Microbiota Transplantation” line 161-164, 203-207, 197-201.
Q8) I suggest adding a table with ongoing clinical trials focused on the role of FMT or probiotics in UC patients to highlight the clinical relevance of this emerging microbiome-based approach.
A8) Thank you for your suggestion. A table of clinical trials registered in clinicaltrial.gov with FMT in UC disease management is now added in the manuscript section 4.1.
Round 2
Reviewer 2 Report
The research is actual. The authors have improved the manusript. The authors use dysbacteriosis as a target for the treatment of UC. However, the ways of influencing the microbiota discussed in the article are limited. Are there other ways to influence the microbiota? I think that this issue is important for the systematization of treatment methods through the impact on dysbacteriosis. There is no answer to the question in this manuscript.
Author Response
Reviewer 2
The research is actual. The authors have improved the manusript. The authors use dysbacteriosis as a target for the treatment of UC. However, the ways of influencing the microbiota discussed in the article are limited. Are there other ways to influence the microbiota? I think that this issue is important for the systematization of treatment methods through the impact on dysbacteriosis. There is no answer to the question in this manuscript.
Thank you for your suggestion. More details regarding other factors that might influencing the dysbacteriosis are now added to the manuscript line 88-98. Furthermore, a systematization of FMT treatment methods, which might impact on dysbacteriosis is now added to manuscript line 230-236

Reviewer 3 Report
I would suggest checking the English and correcting the typos in the Figure legend, e.g. prevelence, incresed, comparsion...
There is an inappropriate use of Capital letters in the main text, please correct it.
Author Response
Reviewer 3
I would suggest checking the English and correcting the typos in the Figure legend, e.g. prevelence, incresed, comparsion...
There is an inappropriate use of Capital letters in the main text, please correct it.
Thank you for your suggestion. The English language in the Figure legend is now corrected by a native English speaker. We have also corrected inappropriate us of the Capital letter.
